# Zhang-Rice singlets state formed by two-step oxidation for triggering water oxidation under *operando* conditions

Chun-Kuo Peng [1], Yu-Chang Lin [1,2], Chao-Lung Chiang [2], Zhengxin Qian[3], Yu-Cheng Huang[4], Chung-Li Dong [4], Jian-Feng Li [3] ✉, Chien-Te Chen[2], Zhiwei Hu [5] ✉, San-Yuan Chen [1] ✉ & Yan-Gu Lin [2] ✉

The production of ecologically compatible fuels by electrochemical water splitting is highly desirable for modern industry. The Zhang-Rice singlet is well known for the superconductivity of high-temperature superconductors cuprate, but is rarely known for an electrochemical catalyst. Herein, we observe two steps of surface reconstruction from initial catalytic inactive $Cu^{1+}$ in hydrogen treated $Cu_2O$ to $Cu^{2+}$ state and further to catalytic active Zhang-Rice singlet state during the oxygen evolution reaction for water splitting. The hydrogen treated $Cu_2O$ catalyst exhibits a superior catalytic activity and stability for water splitting and is an efficient rival of other $3d$-transition-metal catalysts. Multiple *operando* spectroscopies indicate that Zhang-Rice singlet is real active species, since it appears only under oxygen evolution reaction condition. This work provides an insight in developing an electrochemical catalyst from catalytically inactive materials and improves understanding of the mechanism of a Cu-based catalyst for water oxidation.

Understanding how the electrocatalyst/electrolyte interface reforms under operating conditions[1–4] can offer mechanistic insight that allows tracking of the catalytically active motif and enlighten a path towards the development of active and stable electrocatalysts. The nature of such nanoscale interfaces is likely heterogeneous and transforms strongly based on electrochemical operation. To date, however, probing such a dynamic process has been challenging because the surface represents only a tiny fraction of a bulk electrocatalyst, and heterogeneous solid/liquid nanojunctions are nontrivial to characterize in situ[5,6]. Electrochemical water splitting for emerging alternative fuels has been considered one of the most feasible tactics to meet the challenge of decarbonization. Identification of the key intermediate state during the oxygen-evolution reaction (OER)[7–9], which is critical to water splitting and $CO_2$ reduction reactions, remains elusive. The dynamic evolution of irreversible surface reconstruction on an

electrocatalyst during OER and the deciphering of the reaction mechanisms are of utmost importance[6,10,11].

Engineering the coordination environment of a metal center is of fundamental importance in heterogeneous catalysis[12,13]. In particular, modulating the metal–oxygen bonding environment at the electrocatalyst surface offers an effective path toward enhancing the interfacial reactivity. The surface geometric construction and electronic regulation are found to be two decisive factors for the intrinsic improvement of electrocatalytic performance[14]. Accordingly, the first-row $3d$-transition metal (TM) oxides have recently been regarded as promising candidates for OER, owing to their earth abundance and fascinating electronic properties derived from the crystal-field theory. Exemplified by a TM element, the valence variation of a TM plays a pivotal role in catalyzing water oxidation. It is commonly recognized that the high-valent TM species and oxyhydroxides can engender a

[1]Department of Materials Science and Engineering, National Yang Ming Chiao Tung University, Hsinchu 30010, Taiwan. [2]National Synchrotron Radiation Research Center, Hsinchu 30076, Taiwan. [3]State Key Laboratory of Physical Chemistry of Solid Surfaces, iChEM, College of Chemistry and Chemical Engineering, Xiamen University, 361005 Xiamen, China. [4]Department of Physics, Tamkang University, New Taipei City 25137, Taiwan. [5]Max-Planck-Institute for Chemical Physics of Solids, Nöthnitzer Str. 40, Dresden 01187, Germany. ✉e-mail: Li@xmu.edu.cn; Zhiwei.Hu@cpfs.mpg.de; sanyuanchen@nycu.edu.tw; lin.yg@nsrrc.org.tw

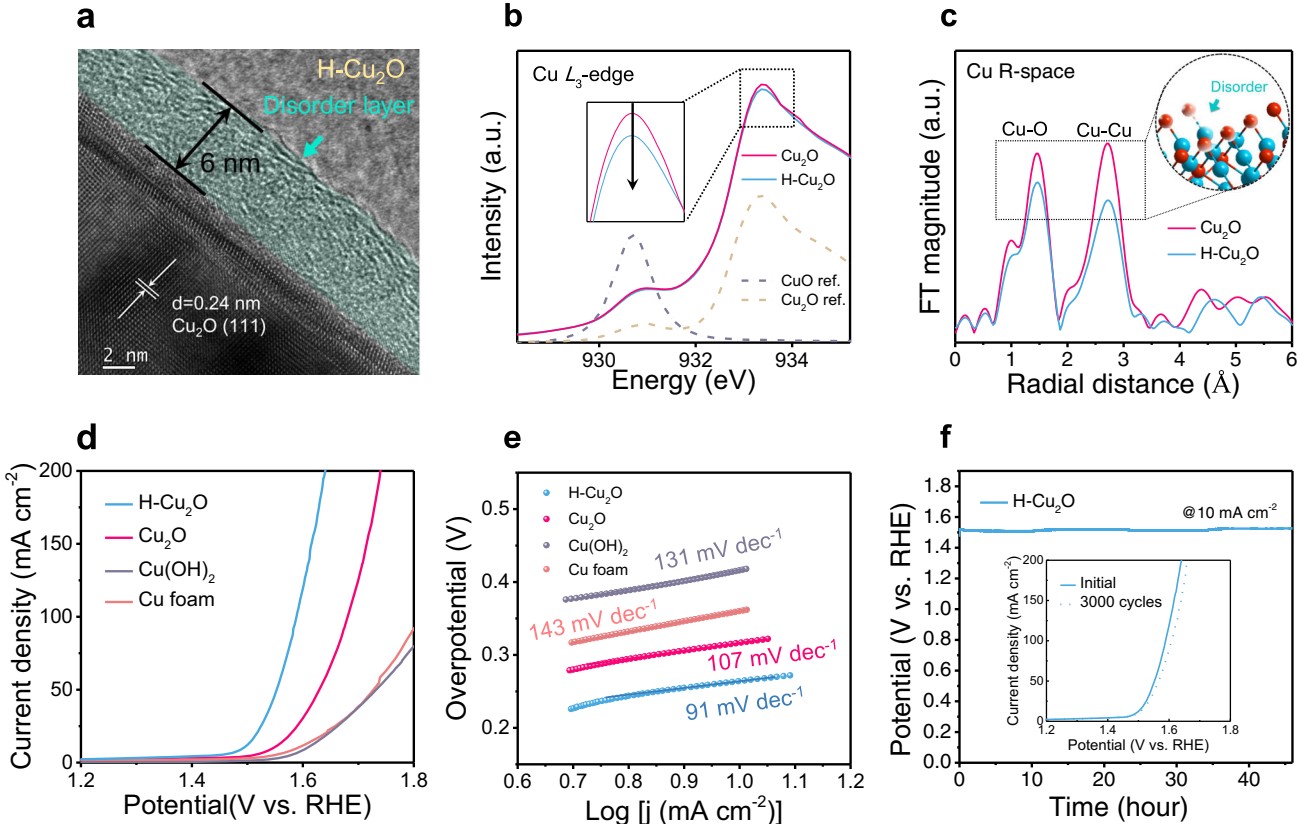

**Fig. 1 | Structural characterization and OER performance of H-Cu₂O catalysts.** **a** TEM image of H-Cu₂O. **b** Cu $L_3$-edge XANES spectra of Cu₂O, H-Cu₂O and references (ref.). **c** Cu $K$-edge EXAFS spectra of Cu₂O and H-Cu₂O. **d** OER polarization curves of Cu₂O, H-Cu₂O, Cu(OH)₂, and Cu foam in 1.0 M KOH solution with 90% iR-correction. **e** The corresponding Tafel plots of all catalysts. **f** The long-term electrochemical stability of H-Cu₂O was measured at current density 10 mA cm⁻² (without iR-correction). The catalyst loadings for Cu₂O and H-Cu₂O were 11 mg cm⁻² and 10.8 mg cm⁻², respectively.

high reactivity towards OER[15]. Accompanying over-oxidation, the strong hybridization between TM $3d$ and O $2p$ facilitates the interaction of electrocatalysts and oxygen-related adsorbates[16], leading to structural disorder and electrochemical irreversibility, which makes tracking technologically challenging. For this reason, it is not straightforward to unravel the nature of the highly covalent bonds in over-oxidized species during OER.

Cu-based materials have been proven to be state-of-the-art catalysts for CO₂ reduction[17,18] and photocatalytic water reduction[19]. So far the OER performance measured with Cu-based electrocatalysts have, however, fallen far short of a satisfactory standard; in particular, they underperform relative to other late $3d$-TM (e.g., Co and Ni) oxides[20,21]. As the cause of this poor performance has not been the focus of extensive studies during the past decade, seeking an inspiring strategy to accelerate OER kinetics on Cu-based electrocatalysts and unveiling their active sites are very challenging but urgent tasks.

Herein, the rational regulation of the active Cu center with unsaturated coordination (denoted H-Cu₂O) via facile hydrogenation is described. Specifically, the substitutional hydrogen induces a surface geometric rearrangement of copper(I) oxide to generate the distinctive oxygen nonstoichiometry, resulting in exceptional performance and durability. Many *operando* spectroscopic tools, including grazing-angle X-ray scattering (GAXS), quick X-ray absorption (quick-XAS), soft X-ray absorption (soft-XAS), Raman spectra and electrochemical impedance spectroscopy (EIS), were utilized to uncover that the Zhang-Rice singlet state[22-25] is unexpectedly observed to participate directly in OER-cycle superconductors. For the first time, a Zhang-Rice singlet state is proposed to trigger electrocatalytic oxygen release. Notably, such a high-valent CuO₄ geometry containing Cu³⁺ with $d^9L$ ($L$: an O $2p$ hole) charged character of Cu $d$-electrons is a paramount key to dominate the oxygen-evolution step of the OER rather than a CuOOH species, showing a fundamental difference from the conventional scheme.

## Results

### Catalyst characterization

A scanning-electron-microscope (SEM) image of the H-Cu₂O catalyst displays a rod-like morphology (Supplementary Fig. 1). This H-Cu₂O catalyst had disordered surfaces, of which the thickness of the disordered outer layer surrounding a crystalline core was about 6 nm as depicted in illustrated in Fig. 1a. A transmission-electron-microscope (TEM) image and the corresponding fast-Fourier-transform (FFT) patterns of the H-Cu₂O catalyst further validate the crystalline core covered with a thin structurally disordered layer, induced by the hydrogenation (Supplementary Fig. 2). Secondary-ion mass-spectrometry (SIMS) depth profiles of the H-Cu₂O catalyst illustrate a clear introduction of hydrogen content in the surface of thickness ca. 6–8 nm (Supplementary Fig. 3). To understand deeply the disordered surface, multiple techniques, including X-ray photoelectron spectra (XPS) and low-grazing-angle quick-XAS, were employed to characterize the samples as prepared. The Cu $2p_{3/2}$ XPS spectra show a main feature at ~932.4 eV[26], corresponding to the chemical state of Cu¹⁺ or Cu⁰ (Supplementary Fig. 4a). The presence of Cu¹⁺ for H-Cu₂O catalyst was confirmed by the dominant signal at 916.8 eV in the Cu LMM spectra (Supplementary Fig. 4b)[26]. As shown in Fig. 1b, soft-XAS at the Cu-$L_3$ edge were employed to probe directly the Cu $3d$ electronic properties of the H-Cu₂O catalysts. The main feature at 933.4 eV in Cu₂O is assigned to the final state $2p_{3/2}p^6d^{10}s^1$ from initial state $3d^{10}$ for Cu¹⁺[27]. Clearly, both of Cu₂O and H-Cu₂O confirm the presence of Cu¹⁺ surface. Since Cu¹⁺ is fully occupied with ten electrons in $3d$, the broad

and weak peak at 933.4 is assigned to the unoccupied of $4s^1$ states[27]. Relative to pristine $Cu_2O$, the decreased intensity at 933.4 eV indicates that the unoccupied states of $4s^1$ decrease in $H-Cu_2O$. This result shows slight charge localization induced by hydrogenation, likely leading to more rapid charge transfer for the OER. X-ray absorption near-edge structure (XANES) and extended X-ray absorption fine structure (EXAFS) were further recorded to reveal the chemical state and local structure of the $H-Cu_2O$ catalysts. Note that there is no obvious energy shift between $Cu_2O$ and $H-Cu_2O$ in the Cu $K$-edge XANES spectra, exhibiting the primary $Cu^{1+}$ features (Supplementary Figs. 5, 6). The result agrees with the Cu LMM Auger spectra (Supplementary Fig. 4b). The Fourier-transformed $k^3$-weighted EXAFS spectra (Fig. 1c), in which two main features at ~1.5 Å and 2.7 Å correspond to the scattering paths of the nearest oxygen (Cu−O) and the secondary copper atoms (Cu−Cu), testify that most coordination sites of the unsaturated Cu centers appear in the $H-Cu_2O$ catalysts. Based on these observations, we believe that the H dopant could reform the local electronic configuration and atomic arrangement of bonded Cu and adjacent O atoms and consequently enhance the localization capacity of charges on Cu atoms. In theory, the bonding strength of oxygen intermediates could depend on the degree of the filling in the antibonding states[28]. The enhanced localization capacity of charges on Cu could contribute to much filling of the antibonding states, resulting in weak adsorption of oxygen intermediates. In contrast, the less filling of the antibonding states would result in the strong adsorption of oxygen intermediates. According to OER volcano plot in metal oxides, the binding strength with oxygen intermediates should be neither too strong nor too weak[29]. For p-type $Cu_2O$, the highest occupied $d$-state is quite closer to the Fermi level, resulting in the less filling of antibonding states and the stronger adsorption of oxygen intermediates. Since the adsorption of oxygen on $Cu_2O$ is too strong that restricts OER activity, the $H-Cu_2O$ is effective for filling much of the antibonding states with weak intermediates adsorption, and thus achieving a better activity[30].

## Electrochemical properties toward OER activity

The OER catalytic performance of the $H-Cu_2O$ catalysts was evaluated using linear-sweep voltammetry (LSV) in KOH solution (1 M) at scan rate $1 \, mV \, s^{-1}$. With a definite coordination-unsaturated structure, the $H-Cu_2O$ model catalysts exhibit a small overpotential 263 mV, small Tafel slope $91 \, mV \, dec^{-1}$ and high durability over 45 h at a current density $10 \, mA \, cm^{-2}$ for OER in alkaline media (Fig. 1d–f). Such an exceptional performance is superior to those of previously reported $3d$ TM-oxide catalysts (Supplementary Fig. 7). To well evaluate the stability of $H-Cu_2O$, the long-term durability at $10 \, mA \, cm^{-2}$ and $100 \, mA \, cm^{-2}$ are tested (Supplementary Figs. 22, 25 and 28). Evidently, the results show negligible decay of $H-Cu_2O$ after OER-100 hours of operation with 93% Faradaic efficiency, indicating good stability of the reconstructed surface. Apart from the overpotential and Tafel slope, the electrochemically active surface area (ECSA), which is estimated from the double-layer capacitances ($C_{dl}$), is another significant controlling factor for the intrinsic activity of catalysts. The $C_{dl}$ value of $H-Cu_2O$ is 13.7% greater than for $Cu_2O$, indicating that hydrogenation certainly augments the number of active sites, which is beneficial for the OER (Supplementary Fig. 8c). To exclude the contribution of larger ECSA for OER performance, a histogram of specific activity of the catalysts with error bar at 1.6 V vs. RHE is given (Supplementary Fig. 8d), reflecting good intrinsic electrocatalytic activity in $H-Cu_2O$ catalysts. We conclude that the improved OER performance of the $H-Cu_2O$ catalysts is attributed to not only the increased ECSA but also the enhanced intrinsic activity owing to the enriched electron densities and coordinatively unsaturated Cu centers. In addition, the $H-Cu_2O$ catalysts disclose a small resistance of charge transfer relative to $Cu_2O$ as extracted from EIS analyses (Supplementary Fig. 9). To probe the OER kinetics and the properties of the catalyst/electrolyte interfaces, we exploited *operando* EIS measurements (Supplementary Fig. 10). As displayed in the 3D contour Bode plots during OER, the phase-angle relaxation at the low-frequency region ($10^0$–$10^1$ Hz) is closely related to the charge transfer at catalyst/electrolyte interfaces. When the applied potential increase, the phase-angle of $H-Cu_2O$ at low-frequency region decreases quickly relative to $Cu_2O$, indicating the superfast OER kinetics due to the low-coordinated environment of Cu atoms.

## Identification of the active site

To capture the dynamic structural reconstruction or transformation of the catalysts, we implemented *operando* GAXS and *operando* quick-XAS during the OER operation. To obtain a more intuitive impression, a customized *operando* liquid electrochemical cell was designed as depicted in Fig. 2a. Figure 2b, c presents 2D contour plots of the color-coded scattering intensities as a function of applied potential for the catalysts recorded with *operando* GAXS at 18 keV of synchrotron X-ray. Stages I, II, and III are assigned to hydroxylation, oxygen evolution and after OER (potential off), respectively. The main characteristic scattering signals are attributed to $Cu_2O$ (111) and $Cu_2O$ (200) facets throughout the entire potential range. In region II, the formation of a $Cu(OH)_2$ phase on the surface of catalysts is captured in both $Cu_2O$ and $H-Cu_2O$ catalysts. Such $Cu(OH)_2$ phases well retain their situation under a potential-off condition, indicating an irreversible surface reformation from $Cu_2O$ to $Cu(OH)_2$ (Fig. 2d). Relative to pure $Cu_2O$, stage II at a much smaller potential (below 1.5 V) for $H-Cu_2O$ catalysts is observed, inferring more rapid deprotonation on the reconstructed surface of $Cu_2O$ catalysts.

*Operando* Cu $K$-edge XANES spectra of the $H-Cu_2O$ catalysts and pristine $Cu_2O$ under the OER are provided in Supplementary Fig. 11. The edge energy of $H-Cu_2O$ at open-circuit potential (OCP) is identical to that of pure $Cu_2O$, confirming the nature of $Cu^{1+}$. Upon increasing the anodic potentials, the absorption edge of Cu $K$-edge XANES spectra for $H-Cu_2O$ rapidly shifts to greater energies relative to pure $Cu_2O$, supporting a more rapid deprotonation on the reconstructed surface of the $H-Cu_2O$ catalysts[31,32]. To probe the electronic structure of the catalysts during OER process, we recorded *operando* Cu $K$-edge XANES spectra. The absorption edge (the first-derivative signal) denoted with a dotted line in Fig. 3a, b, shows a smaller energy shift with increased applied voltage, indicating an increased average Cu valence state. A sharp shift was located at a smaller potential (1.4 V) for $H-Cu_2O$ catalysts vs. 1.6 V for $Cu_2O$. Besides, the appearance of a slight shoulder at ~8985 eV on the rising $K$-edge XAS at applied voltage 1.5 V for $H-Cu_2O$ and 1.7 V for $Cu_2O$, respectively. The feature can be assigned to four-coordinated square–planar geometry[33]. On this basis, the oxidation state of the copper site at stage II might be assigned as +3, showing the electrochemically driven conversion of $Cu(OH)_2$ into $CuO_4$ geometry probably. We recorded also the *operando* Cu $K$-edge $k^3$-weighted EXAFS spectra, which are sensitive to the local crystal structures of an absorbing metal ion. The fitted profiles of Fourier-transformed EXAFS spectra (FT-EXAFS) as results are presented in Supplementary Figs. 12 and 13 and Supplementary Tables 1 and 2, respectively. As shown in Fig. 2e, f, the corresponding 3D FT patterns along with 2D contour plots reveal a remarkable two-step dynamic structural evolution during electrochemical operation. The detailed coordination environment involving the relative coordination numbers and bond lengths of Cu centers is summarized quantitatively in Fig. 3c and Supplementary Fig. 14. Before the oxygen evolution (i.e., region I), the coordination numbers ($\Delta N/N_{OCP}$) and bond lengths ($\Delta R/R_{OCP}$) of Cu−O for $H-Cu_2O$ catalysts sharply increased when the applied potential was greater than OCP. For $H-Cu_2O$ catalysts, the largest $\Delta N/N_{OCP}$ and $\Delta R/R_{OCP}$ were obtained at the onset of OER at 1.4 V, because of a phase transition from $Cu_2O$ to $Cu(OH)_2$ (Fig. 3c). In contrast, $\Delta N/N_{OCP}$ and $\Delta R/R_{OCP}$ of pure $Cu_2O$ slowly increased with applied potential up to 1.6 V (Fig. 3d). Note that the hydrogenation treatment is an effective route to accelerate Cu pre-oxidation and surface reconstruction during OER. As the applied potential increased

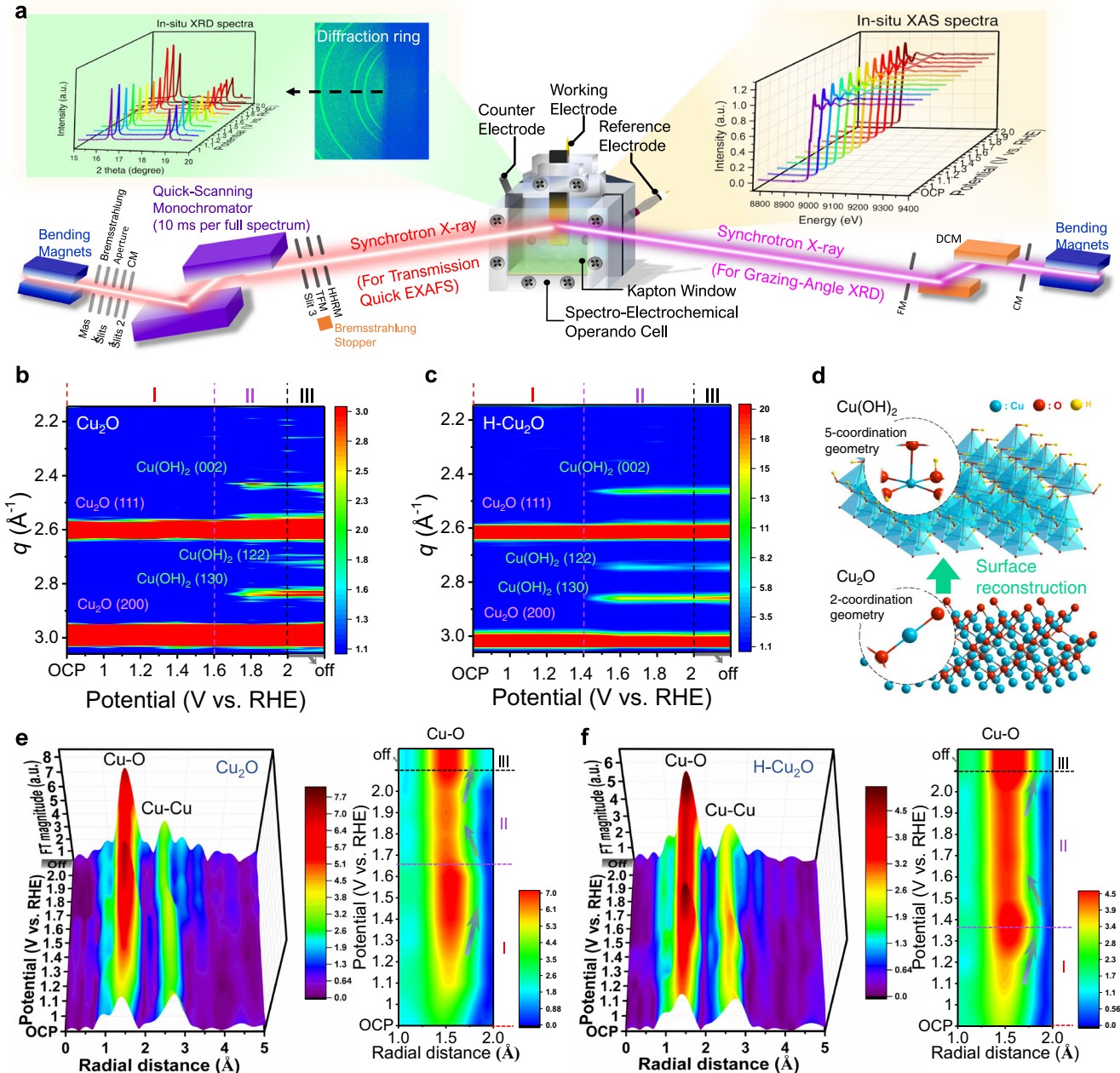

**Fig. 2 | *Operando* GAXS and quick-XAS characterizations for the OER.**
**a** Schematic illustration of the *operando* liquid electrochemical cell for the GAXS and quick-XAS apparatus. *Operando* GAXS of (**b**) pure $Cu_2O$ and (**c**) H-$Cu_2O$ catalysts. **d** Proposed structural reconstruction during OER. *Operando* Fourier-transformed EXAFS of (**e**) pure $Cu_2O$ and (**f**) H-$Cu_2O$ catalysts (stages I, II and III are assigned to hydroxylation, oxygen evolution and after OER (potential off), respectively).

further to 1.6 V (i.e., region II), $\Delta N/N_{OCP}$ and $\Delta R/R_{OCP}$ of the catalysts decreased gradually, likely implying the electrochemical oxidation of $Cu(OH)_2$ to form an over-oxidized Cu species during oxygen evolution (Supplementary Fig. 14). This is also accordant with the *operando* XANES spectra (Fig. 3a, b). Such structural reconstruction of Cu species for H-$Cu_2O$ catalysts is also evident from the wavelet transform (WT) analyses of EXAFS spectra as depicted in Fig. 3e. Two dashed lines at 5.6 and 7.7 Å$^{-1}$ are defined as the strongest oscillation amplitudes of Cu–O and Cu–Cu bonds. The yellow region represents the FT magnitude of the Cu–O and Cu–Cu bonds in H-$Cu_2O$ during OER and two references located about 1.8 and 2.7 Å. In the OCP state, the coordination environment of Cu in H-$Cu_2O$ was confirmed as $Cu_2O$ because of a similar pattern. On switching on a voltage, a phase transition from H-$Cu_2O$ toward $Cu(OH)_2$ occurred (stage I).

Upon further increased applied voltage from 1.4 V, $Cu(OH)_2$ was further transformed to a *operando*-generated copper species, which has different patterns for Cu–O and Cu–Cu bonds in stage II. In addition, the Cu–Cu bond shows a different amplitude distribution between stages I and II in the *k*-space from 7.7 to 6.1 Å$^{-1}$. After OER, the pattern of the Cu–Cu bond shifted to a larger *k* value and a larger Cu–Cu bond distance; phase III was formed. The surface chemistry of H-$Cu_2O$ catalysts was hence notably altered by dynamic surface reconstruction during OER. This effect indicates that the reconstruction facilitated by hydrogenation is the vital key to evolve oxygen effectively.

The electronic structures of the real active sites in H-$Cu_2O$ catalysts during the OER were well assessed using *operando* soft-XAS (Fig. 4) at the Cu $L_3$-edge and O $K$-edge as known from previous

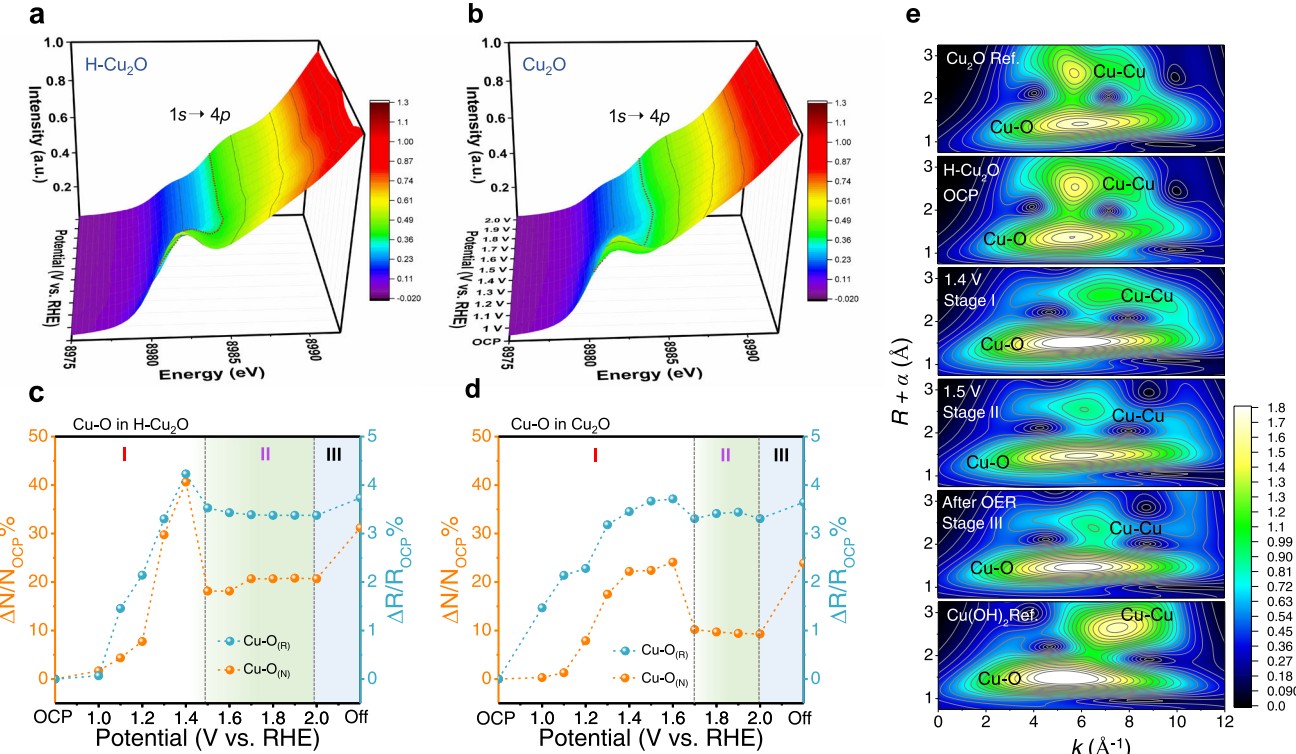

**Fig. 3 | Understanding the correlation between OER activity and local structure transformation. a, b** 3D patterns of *operando* Cu *K*-edge XANES of H-Cu₂O and Cu₂O catalysts. **c, d** Structural coherence change of Cu−O in EXAFS coordination number (N) and bond length (R) of H-Cu₂O and Cu₂O catalysts under an applied potential relative to the OCP state. **e** Comparison of Cu *K*-edge WT-EXAFS recorded for H-Cu₂O, standard references (Ref.) and catalytic materials at OCP, 1.4, 1.5 V and after OER (stages I, II, and III are assigned to hydroxylation, oxygen evolution and after OER (potential off), respectively).

studies on the high-temperature superconductors (high-$T_c$) Cu superconductors[34,35] and even more complicated Cu oxide[36,37]. The major challenge is to separate the liquid from the ultrahigh vacuum for OER experiments. Figure 4a shows an *operando* flow cell with an ultrathin Au@Si₃N₄ membrane window to separate the liquid cell from the ultrahigh vacuum condition. Figure 4b shows the Cu-$L_3$ XAS of H-Cu₂O in regions I−III. At the beginning of stage I (OCP state), we see a broad feature at 933.4 eV (black circles) for an initial Cu¹⁺ ($3d^{10}$) state. A sharp feature at 931.3 eV appeared at applied voltage 1.4 V in region II (green circles) and is assigned to Cu²⁺ ($2p^5 3d^{10}$ final state from $3d^9$ initial state) from Cu(OH)₂. Most interestingly, a high broad shoulder feature at 932.4 eV appeared under the OER condition, which is well known in the observed La₁₋ₓSrₓCuO₄ (Supplementary Fig. 15) corresponding to the doped hole for Cu³⁺ state with $2p^5 3d^{10}L$ final state[34,35,38] ($L$ denotes a hole in the O $2p$ states). This completely agrees with the *operando* XANES results (Fig. 3a, b). The spectral weight of this final state is weak as doped holes are located mainly at the O $2p$ state[19,38]. Upon switching off the applied voltage, this Cu³⁺ state immediately vanished and only Cu²⁺ remained. As the doped holes are located mainly at the O $2p$ states, the quantitative content of the related state Cu³⁺ is expected to be observed from the O-*K* XAS spectra. It is well known that for a charge-transfer system with increased valence state, the pre-edge features shift to lower energy, and their spectral weight increases[35,39].

As shown in Fig. 4c, a strong pre-edge peak at 530.3 eV occurs under an applied voltage below 1.4 V, which is assigned to the transitions to the upper Hubbard band from the O $1s$ core level (corresponding to the Cu²⁺ state for simplicity). Under OER conditions (stage II) a feature below 529 eV was observed, which is attributable to transitions from O$1s$ to the doped hole states constructed by the strong O $2p$-Cu $3d$ hybridization so-called Zhang-Rice singlet state or Cu³⁺ state[35]. A similar feature was recently observed also for cuprate superconductor Ba₂CuO₄₋ᵧ, wherein the local octahedron is in an

exceptionally compressed version[40]. Both the Cu-$L_3$ and the O-*K* XAS spectra hence demonstrate the existence of a Zhang-Rice singlet state or Cu³⁺ state under OER conditions. The Cu³⁺ ($3d^9L$) states are split into two poorly resolved peaks at 528.17 eV and 529 eV. A similar splitting was observed in Sr₁₄Cu₂₄O₄₁ originating from a different local environment of oxygen ions[36,41] (Supplementary Fig. 16c). Note that the Zhang-Rice singlet state disappeared upon switching off the applied voltage, which means that this real OER active species cannot be observed in experiments ex situ.

In addition to the Cu³⁺ species, a feature at 531.4 eV appears, which can be assigned to a CuOₓ(OH)ᵧ-related intermediate[1]. After switching off the applied voltage, those features disappear. To obtain the detailed spectral weight of the Zhang-Rice singlet-state-related spectral weight under OER conditions, we analyzed the O-*K* XAS (black circles) after subtracting an edge jump (black line in Fig. 4c) in the same way as used for high-$T_c$ Cu superconductors[34] as shown in Supplementary Figs. 16a and 17 with the O-*K* XAS of La₂₋ₓSrₓCuO₄[34] (Supplementary Fig. 16b) for comparison. The relative spectral weight of Cu³⁺ species corresponds to $x = 0.1$ in La₂₋ₓSrₓCuO₄.

## Proposed catalytic mechanism

Based on all above *operando* X-ray spectroscopic data, a schematic electronic structure at 1.4 V and under OER conditions is depicted in Fig. 4d. As a first step, Cu₂O with Cu¹⁺ ($3d^{10}$) state transfers gradually to Cu(OH)₂ below 1.5 V in the region I. The Cu²⁺ ion in the Cu(OH)₂ phase has a five-coordinated square-pyramidal geometry. On further increasing the applied voltage, part of Cu²⁺ was transferred to Cu³⁺ ($3d^9L$) in CuO₄ with four-coordinated square−planar geometry in stage II. After switching off the applied voltage, the Cu³⁺ ($3d^9L$) species returned to the Cu(OH)₂ state. With the valence state of the late TM ion increased by one unit, the charge transfer energy decreased by 3−4 eV and even became a negative value. The O $2p$ character gained weight

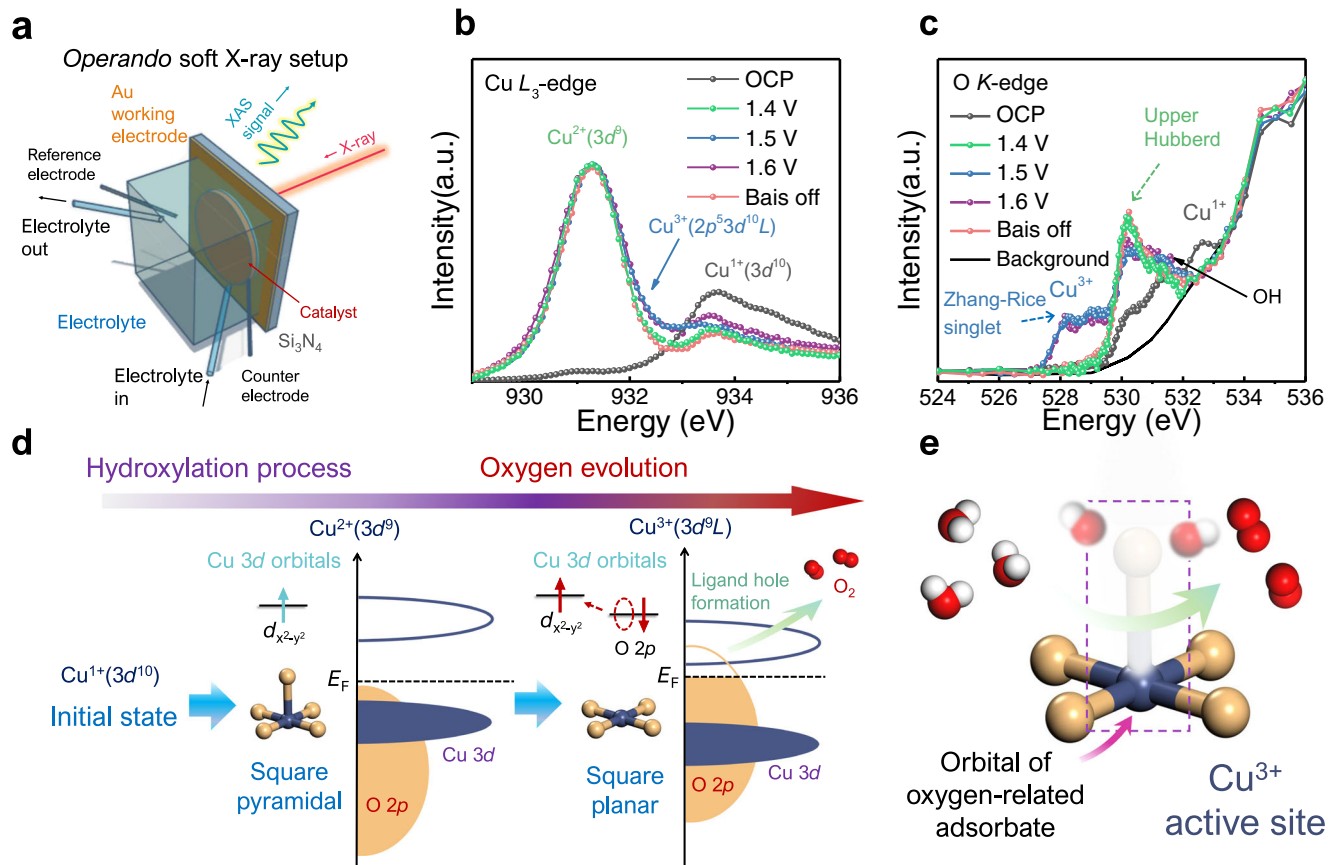

**Fig. 4 | *Operando* soft-XAS characterization of high-valent Cu as an active site during OER. a** Schematic illustration of *operando* soft-XAS setup under ultrahigh vacuum condition. **b** Cu $L_3$-edge and **c** O $K$-edge of H-Cu$_2$O. **d** Electronic spin states and orbital physics of Cu site in OER. **e** Schematic illustration of copper geometry site interaction with oxygen-related adsorbate during oxygen evolution.

above $E_F$ and shifted near $E_F$[42,43], which means that the covalence between Cu 3$d$ and O 2$p$ increased from Cu$^{1+}$ to Cu$^{2+}$ and further to a Cu$^{3+}$ state. Previous work indicated that the catalytic activity of TM oxides could be enhanced on increasing the covalence between TM 3$d$ and oxygen 2$p$ orbitals[6]. Figure 4e shows the oxygen generation from the surface of H-Cu$_2$O at the Cu$^{3+}$ (3$d^9L$) state, which had a square–planar geometry with a bare Cu$^{3+}$ as an active site. The CV analyses for catalysts at scan rate of 5 mV s$^{-1}$ are supplemented to support the in situ generation of Cu$^{3+}$ species as shown in Supplementary Fig. 26. During the oxidation process, only one anodic peak appeared at low potential and was attributed to the conversion of Cu(I) into Cu(II). Coincidentally, the anodic peak of Cu(II)/Cu(III) oxidation at the high- potential region highly overlapped the large OER current response. Even so, in the subsequent reduction process on the reverse potential scan, the broad cathodic peak was observed and resulted from the reductive transformation of Cu(III) to Cu(II).

To explore further the unique geometry sites, we recorded *operando* Raman spectra of H-Cu$_2$O in KOH (1.0 M). The Raman spectra in Supplementary Fig. 18 show a comparison of H-Cu$_2$O and references. As shown in Supplementary Fig. 19a, the spectra also exhibited two major parts, stage I (hydroxylation below 1.5 V) and II (oxygen evolution above 1.5 V). The H-Cu$_2$O phase transition was gradual from Cu$_2$O to Cu(OH)$_2$ with increasing potential in stage I. Three peaks appeared at 292.2, 490.1, and 3557.4 cm$^{-1}$, of which the latter corresponds well to the stretching vibration of O–H in Cu(OH)$_2$ reference. In stage II, a peak at 603.9 cm$^{-1}$ appeared, due mainly to the framework vibration of Cu–O in CuO$_4$[44]. In the deuterium water experiment (Supplementary Fig. 19b), the peak at 490.1 cm$^{-1}$ obviously shifted to 480.1 cm$^{-1}$ with a shift ratio of 97.9%, which further proves that the 480.1 cm$^{-1}$ peak is

attributed to the Cu–OH structure. Furthermore, the redshift of 3557.4 cm$^{-1}$ to 2628.8 cm$^{-1}$ verifies the existence of the O–H structure. It must be clarified that there is no obvious redshift at 605.1 cm$^{-1}$ in the deuterium experiment, indicating that there is no H in this structure. In addition, the potential is also relevant to the appearance and disappearance at of the peak at 605 cm$^{-1}$ (Supplementary Fig. 20). The structure CuO$_4$ disappeared quickly when the potential was cut off. It demonstrates that CuO$_4$ is the active structure; this result is consistent with previous X-ray measurements. A continuous phase transition can also be demonstrated in the TEM images of H-Cu$_2$O catalyst as shown in (Supplementary Fig. 21). After the OER catalytic process, the H-Cu$_2$O crystal structure consisted of a Cu$_2$O core (circle I) and a Cu(OH)$_2$ shell (circle II) (Supplementary Fig. 21a). The crystal planes related to the Cu$_2$O core and Cu(OH)$_2$ shell were also identified in SAED patterns, which is consistent with previous GAXS patterns (Supplementary Fig. 21b, c). The average depth (~30 nm) of the surface layer of outer Cu(OH)$_2$ was further observed on an enlarged scale of TEM images (Supplementary Fig. 21d–f), but the reaction depth (~30 nm) is much larger than a disorder layer (~6 nm) as such reconstruction occurred also for a pure Cu$_2$O phase. Another interesting finding is the further generation of an amorphous region on the catalyst surface. The amorphous region has putative active sites that are tuned back from the oxygen-related species during the OER. These *operando* X-ray measurements and TEM results indicate that the structural disorder led to the rapid surface reconstruction at H-Cu$_2$O in OER[31,45]. Taking into account all complementary information from multiple *operando* experiments including GAXS, quick-XAS, soft-XAS, Raman, and EIS, the active-site configuration and reaction cycle are proposed in Fig. 5, taking advantage of the fact that OH$^-$ is preferentially adsorbed on the

**Fig. 5 | Proposed OER mechanism for H-Cu₂O.** Dynamic configuration of active sites during OER.

region of the coordinative unsaturation, and additionally the intrinsic tendency of O 2$p$ to favor the delocalization of local electrons for a noticeable transformation of the pre-designed metal–oxygen bonding environment. The impact of the hydrogenation on H-Cu₂O is observed herein to generate the coordinatively unsaturated Cu centers with strong charge localization, which allows the Cu cations to be easily over-oxidized, thereby leading to a facile transformation to high-valent CuO₄ geometry[44]. It is noticeable that we report on *operando* spectroscopic observations of the Zhang-Rice singlet state responsible for managing the oxygen-evolution step in the form of high-valent Cu$^{3+}$ with $d^9L$ configurations as revealed by soft-XAS. To the best of our knowledge, a Zhang-Rice singlet state serving as active center has not yet been observed for OER electrocatalysts to date. Our results provide direct evidence of high-valent CuO₄ sites, rather than oxyhydroxide species, as the key intermediate state of the pre-equilibrium step on H-Cu₂O for oxygen evolution. Furthermore, as shown in Supplementary Fig. S23, the H-Cu₂O catalyst exhibits pH-dependent OER activity, implying that non-concerted proton-electron transfers may participate in catalyzing the OER[46]. Under alkaline conditions, electrochemically driven deprotonation results in the intramolecular hydroxyl nucleophilic attack pathway where the adjacent OH⁻ attacks CuO₄ to form the O–O bond. Therefore, a pH-dependent nucleophilic attack pathway for O–O bond formation might be presented as shown in Supplementary Fig. S24.

As demonstrated for H-Cu₂O electrocatalysts, several *operando* spectral methods allowed us to disentangle the dynamic restructuring during OER at nanoscopic solid/liquid interfaces. This report is the first of the unexpected observation of the Zhang-Rice physics responsible for managing the oxygen-evolution step in the form of high-valent Cu$^{3+}$ with $d^9L$ charge character. Specifically, maximizing the hybridization between Cu(3$d$) and O(2$p$) states with the additional appearance of a ligand hole in O 2$p$ orbitals favors oxygen-evolution catalysis involved in the OER cycle. Notably, the nature of the definite Zhang-Rice singlet state corroborates the high-valent CuO₄ geometry with four-coordinated square–planar geometry that served as the key intermediate state in the pre-equilibrium step. As far as we are aware, this unusual observation is in sharp contrast with the commonly proposed scheme, predicting an oxyhydroxide species as the active center for OER electrocatalysts. Our work emphasizes that the charge and spin states of TM oxides would be essential to catalyze oxygen evolution during water oxidation.

## Methods

### Preparation of pristine Cu₂O and H-Cu₂O electrodes
The electrodeposition of Cu₂O was implemented in a two-electrode configuration, including working (Cu foam) and counter (platinum foil) electrodes. The Cu foam was cleaned with sequential ultrasonic treatments in hydrochloric acid (1 M) and deionized water for 10 min each. The surficial Cu(OH)₂ layer on the Cu foam was formed by the subsequent anodization in a NaOH solution (1 M). The electrodeposition condition was performed under a constant current density 4.5 mA cm$^{-2}$ at 18 °C until the potential 1.6 V was reached. These Cu₂O electrodes with surficial Cu(OH)₂ layers were rinsed with copious water, dried in air and annealed at 500 °C (heating rate 5 °C min$^{-1}$) in a tubular furnace with a flowing nitrogen stream for 5 h.

### Preparation of H-Cu₂O electrodes
In a typical synthetic procedure of a hydrogenation treatment, the H-Cu₂O electrode was set up based on pristine Cu₂O on applying gaseous H₂ (99.999%) as the hydrogen source. First, the pristine Cu₂O electrode was placed in a tubular silica furnace and kept under vacuum for 1 h. The furnace was then filled with hydrogen near room temperature. After annealing at 100 °C for 36 h under 1.5 bar H₂ atmosphere we obtained H-Cu₂O electrodes. The catalyst loadings for Cu₂O and H-Cu₂O were 11 mg cm$^{-2}$ and 10.8 mg cm$^{-2}$, respectively.

### Characterization
The morphology of the catalysts was characterized using SEM (JEOL, JSM-6700F) and TEM (JEOL, ARM-200FTH). The high-resolution XPS were measured at TLS beamline BL-24A of the National Synchrotron Radiation Research Center (NSRRC), Taiwan. The XPS measurements were performed under ultrahigh vacuum condition (<10$^{-6}$ bar). The binding energies of collected spectra were calibrated to Au 4$f_{7/2}$ of 84 eV for comparison. The XANES measurements were performed at Taiwan beamline BL-12XU in Spring-8, Japan. Athena software was used to process the acquired EXAFS data. The soft-XAS was collected in fluorescence mode at TLS beamline BL-11A of NSRRC, Taiwan.

### Electrochemical measurements
All electrochemical measurements were made out at room temperature in a typical three-electrode system with an electrochemical potentiostat (CHI 6278E, CHI Instruments). The counter and reference electrodes were platinum foil and Hg/HgO, respectively. All applied potentials were calibrated to a reversible hydrogen electrode (RHE, $E_{RHE}$) for comparison, as shown in Equation. (1) $E_{RHE} = E_{Ref} + 0.059\,pH + E_{Exp}$. ($E_{RHE}$ (V): the potential of reversible hydrogen electrode; $E_{Ref}$ (V): the potential of reference electrode; $E_{Exp}$ (V): the potential of working electrode). For OER measurements, cathodic linear-sweep voltammetry with scan rate 1 mV s$^{-1}$ was performed in KOH (1.0 M, pH -13.6). The 90% iR compensation was applied for OER test by using the automated iR-correction function of the potentiostat. Gas chromatography (GC) was used to determine the Faradaic efficiency.

The ECSA was estimated from the electrochemical double layers of the catalyst surface using Equations. (2) $C_{dl} = i_v/(dE/dt)$ and (3) ECSA = $C_{dl}/C_S$. (where $C_{dl}$ is the double-layer capacitance and dE/dt is the scan rate.) The potential windows were measured from 1.1 to 1.2 V versus RHE. The geometric areas of the electrodes were calculated on

dividing the general specific capacitances of $C_s = 40\,\mu F\,cm^{-2}$ for TM oxide in alkaline solution[47].

## *Operando* EIS measurements
The EIS were performed in a frequency range from $10^{-1}$ to $10^4$ Hz with a small AC amplitude, 10 mV, under applied potential range from 1 to 2 V versus RHE.

## *Operando* GAXS measurements
The GAXS was performed at beamline BL12-B2 of SPring-8 in Hyogo, Japan. The GAXS patterns were collected with a large Debye–Scherrer camera. The scattering angle was aligned to the Bragg peak of standard $CeO_2$ powder (SRM 674b). The incident grazing angle of the X-ray was set at 1°; the X-ray energy was 15 keV ($\lambda = 0.82656$ Å). *Operando* GAXS measurements were performed in a self-assembled Teflon cell sealed with a Kapton tape window ($2 \times 2$ cm$^2$) that was similar to the three-electrode electrochemical condition. The applied potential on the electrode was measured from 1.0 to 2.0 V versus RHE with CHI Instruments. The incident X-ray beam was transmitted through the Kapton window and the electrolyte to collect the surficial GAXS pattern of the electrode.

## *Operando* quick-XAS measurements
The Cu $K$-edge XAS were measured at the TPS beamline BL-44A in NSRRC, Taiwan. *Operando* quick-XAS measurements were performed in the aforementioned self-assembled Teflon cell. The X-ray beam was transmitted through the Kapton tape and electrolyte and reached the detector for XAS collection in the transmission mode.

## *Operando* soft-XAS measurements
The Cu $L_3$-edge XAS measurements were performed at the photoemission end-station at beamline BL-11A in NSRRC, Taiwan. *Operando* soft-XAS were also recorded in a three-electrode setup with the previous self-assembled cell. A gold-covered $Si_3N_4$ window was in contact with copper wires as the working electrode; Hg/HgO and platinum wires were respectively used as reference and counter electrodes. The catalyst powders were dispersed in ethanol with Nafion solution (20 μL, 5%, Sigma-Aldrich), and then sonicated for 10 min. The catalyst ink was drop-cast onto the gold-covered $Si_3N_4$ window. The X-ray beam was transmitted through the $Si_3N_4$ window and reached the detector for soft-XAS spectra collection in the fluorescence mode.

## Data availability
The raw data generated in this study have been deposited in the figshare Public Data Repository database under https://doi.org/10.6084/m9.figshare.21890511. Source data are provided with this paper.

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

## Acknowledgements

Y.G.L. acknowledges support by Taiwan Ministry of Science and Technology (grants MOST 108–2112-M-213-002-MY3, 111-2634-F-A49-007 and 111–2112-M-213-012). Y.G.L. acknowledges support from Taiwan National Synchrotron Radiation Research Center. Y.G.L. also acknowledges the support by the Center for Emergent Functional Matter Science of National Yang Ming Chiao Tung University from The Featured Areas Research Center Program within the framework of the Higher Education Sprout Project by Taiwan Ministry of Education (MOE). We also acknowledge the support from the Max Plank-POSTECH-Hsinchu Center for Complex Phase Materials.

## Author contributions

C.K.P. and Y.G.L. conceived and designed the project. C.K.P. contributed to the sample preparation and analyzed the overall experimental data under the supervision of S.Y.C. and Y.G.L. Y.C.L. and C.L.C. supported *operando* hard XAS experiments. Y.C.H. and C.L.D. supported the liquid cell for soft-XAS experiments. C.K.P, C.T.C., and Z.W.H. contributed to soft-XAS experiments and data analyses. Z.X.Q. and J.F.L. performed the *operando* Raman experiments. C.K.P. and Y.G.L. wrote the manuscript. All the other authors discussed the results and assisted during the manuscript preparation. Y.G.L. was responsible for project management.

## Competing interests

The authors declare no competing interests.
