## [Peer review file · Nature Communications]

REVIEWER COMMENTS

Reviewer #1 (Remarks to the Author):

Applying decent operando spectroscopy characterizations, Peng et al. have revealed every interesting mechanistic insight into the active structure for Cu-based electrocatalysts under OER condition. The concept is new, experimental evidence is solid, but the mechanism presentation is weak at this stage and some other issues also need to be addressed before publication.

It is good to see good OER performance in Cu-based materials. However, one big issue for Cu-based material has been the possible corrosion under strong alkaline and high potential according to the Pourbaix diagram. To understand the stability of the structure better, it is good to have ICP data on the electrolyte and see how the catalyst is dissolved or corrosive under alkaline OER condition.

The Cu₂O catalysts have been prepared firstly by anode electrodeposition with the Cu(OH)₂ as the precursor, and the annealing. According to the current demonstration, the as-synthesized Cu₂O would eventually evolve into Cu(OH)₂ again. The authors need to explain why a transformation from Cu(OH)₂ to Cu₂O by annealing is needed. Also, the precursor Cu(OH)₂ should also be one of the control sample. In Methods, the description for the synthesis is confusion, such as “surficial Cu(OH)₂ layer on Cu₂O electrodes”, maybe it is on the Cu foam?

In Fig. 5, the structure of CuO₄ square-planer is very confusing, since now it sites outside the OER cycle, which suggest it not one the intermediates for OER. Also now two Cu centers are included in the OER cycle but only one is involved. For the active-site configuration and reaction cycle, more details still need to be double-checked. It is very clear that the CuO₄ square-planer is accompanied with OER, it is very promising to be the active structure. However, there is still some possibility that it is not. To avoid the unnecessary mistake, maybe the author could soften the claims a little bit. No matter what it is, it is still needed to put this CuO₄ square-planer into the OER cycle more properly. If possible, please also try to label all the Cu oxidation state at the OER cycle.

According Fig. 5, the conventional adsorbed evolution mechanism and intermolecular nucleophilic attack have been proposed for these Cu-based electrocatalysts. If it is the case, then there should be some special pH-dependence properties. To strengthen these points, the pH-dependence results should be provided.

In Figure 2d, the surface reconstruction is driven by applied potential so it is not “self-reconstruction”.

In Supplementary Figure 7, it is not a fair comparison since different substrates are used, such herein it is Cu foam, others could be carbon cloth or glassy carbon.

Reviewer #2 (Remarks to the Author):

The authors reported new OER catalyst H-Cu₂O exhibits a superior catalytic activity and stability compared with other Cu-based OER catalysts, which possessed two steps of surface reconstruction from initial catalytic inactive Cu¹⁺ in H-Cu₂O to Cu²⁺ state and further to catalytic active Zhan-Rice singlet state, which is demonstrated as the real active sites for OER rather than CuOOH. This work provides a new insight in understanding of the mechanism of a Cu-based catalyst for water oxidation. Even though the manuscript displayed attractive proofs for Zhang-Rice singlet, some related experiments and discussions are needed to be supplemented before accepted in Nature communications. The detailed comments are as follows:

1. In line 100, the authors claimed that the main feature at 933.4 eV in Cu₂O is assigned to final state $1s3d104s1$ from initial state $3d10$ for Cu¹⁺, while it was pointed out that the feature was originated from $2p3/2p6d10s1$ in ref 27. The authors are suggested to confirm the results.
2. In line 101, the authors claimed that the relative lower intensity at 933.4 eV indicated that the unoccupied states of $4s1$ increase in H-Cu₂O, which should be explained and demonstrated by some related literature.
3. The oxygen vacancies in Figure 1c should be confirmed by O 1s spectra or electron paramagnetic resonance (EPR) or Positron annihilation lifetime spectrum (PALS). Besides, the existence of oxygen vacancies generally resulted in lower valence state while the Cu in H-Cu₂O still was Cu¹⁺, which is quite weird. The authors are suggested to explain the phenomenon.
4. LSV curves after 10, 20-, 30-, 40-, and 50-hours durability test at 10 mA cm⁻² should be supplemented to support the stability of the reconstructed surface. Besides, to well evaluate the stability of H-Cu₂O, the durability at 100 mA cm⁻² should be tested for 100 hours.
5. In line 113, the authors held the view that the enhanced localization capacity of charges on Cu atoms probably weakened the chemisorbed oxygen intermediates and improved the water oxidation performance, which should be well discussed the relationship between the localization capacity of charges and oxygen intermediates adsorption. In addition, some related literatures on the binding strength with oxygen intermediates for pure Cu₂O should be well cited and discussed to demonstrate that the weakened chemisorbed oxygen intermediates of Cu₂O are beneficial to OER performance.
6. CV analysis at 5 mV/s in wide potential range should be added to display the redox peaks of Cu^I to Cu^{II} and Cu^{II} to Cu^{III} to support the in-situ generation of Cu^{III} species, which would be consistent with the Figure 2a theoretically.
7. The Tafel region should be reselected in current density between 5 and 10 mA cm⁻² in Figure 1e.
8. In line 191, the authors claimed that “the Cu-Cu bond shows a different amplitude distribution between stages I and II in the k-space from 7.7 to 6.1 Å⁻¹.” However, in Figure 3e at 1.5 V of stage II, the so-called “Cu-Cu bond” possessed almost the same k-space value as Cu-O bond, which should be ascribed to Cu-O bond in second shell. The authors are suggested to Re-discuss the wavelet transform section.

9. It should be stated clearly that the data of “during OER” in Figure 4b and 4c was collected at what OER potential and more XAS data at different OER potentials such as 1.6 V and 1.8 V should be supplemented to support the Zhang-Rice singlet was the active sites for OER in H-Cu₂O.

10. The Zhang-Rice singlet was observed in H-Cu₂O as the reconstruction surface Cu(OH)₂ generation. Herein, it should be pointed out whether Zhang-Rice singlet can be observed for pure Cu(OH)₂ or pure Cu₂O.

11. The improved activity of H-Cu₂O was ascribed to the formation of disordered layer compared with pure Cu₂O. However, the activity comparison between H-Cu₂O and pure Cu₂O after durability test at 100 mA cm⁻² for 100 hours should be supplemented since both H-Cu₂O and pure Cu₂O finally reconstructed Cu(OH)₂ on the surface.

Reviewer #3 (Remarks to the Author):

The manuscript by Peng et al. provides a thorough study of Cu-based catalysts for the oxygen evolution reaction (OER). In spite of the vast variety of operando, in situ and ex situ methods, the analysis and discussion of the obtained results is not fully provided thus it is hard for a reader to follow the logic and make any conclusions based on the presented information.

First of all, I suggest that this work is published as a full paper (not Nature Communication) where the authors should properly discuss all their results.

Secondly, the following changes are suggested to the authors:

1) The authors included a lot of data in the graphs which are hard to read and analyze. E.g. XANES data contain all the measured potentials. It would be more beneficial to select the critical potentials and plot them together for Cu₂O and H-Cu₂O (where the full set remains in SI). Moreover, please add the XANES spectra for the reference samples

2) The discussion of Cu K edge spectra (as well as other techniques, see my main comment above) is very limited. The authors discuss only the edge shift and its correlation with oxidation state. The discussion of the pre-edge region is absent, where this part of the XANES spectra contains vital information on the local coordination and should be correlated with EXAFS and Cu L edge presented later. The authors claim the formation of square planar Cu sites under OER conditions, however, this should be clearly seen in the Cu K edge spectra (pre-edge region, see e.g. <https://pubs.acs.org/doi/10.1021/acs.jpcclett.8b00675>). The conclusions on the geometry changes upon the reaction should be reconsidered based on the available data.

3) Please add Cu L edge data recorded on Cu₂O sample (in situ if possible) and the references for comparison.

Thank you for reviewers' serious and constructive comments on our manuscript. According to their suggestions, the manuscript has been revised as a letter to editor. The revisions and replies we have made are as follows:

Reviewer: 1

Applying decent operando spectroscopy characterizations, Peng et al. have revealed every interesting mechanistic insight into the active structure for Cu-based electrocatalysts under OER condition. The concept is new, experimental evidence is solid, but the mechanism presentation is weak at this stage and some other issues also need to be addressed before publication.

We thank the reviewer for positive statements and suggestion to improve our manuscripts.

Q1 : It is good to see good OER performance in Cu-based materials. However, one big issue for Cu-based material has been the possible corrosion under strong alkaline and high potential according to the Pourbaix diagram. To understand the stability of the structure better, it is good to have ICP data on the electrolyte and see how the catalyst is dissolved or corrosive under alkaline OER condition.

Reply :

Many thanks for reviewer's helpful comments on our manuscript.

According to the referee's sincere and crucial suggestions, the ICP-MS analyses of the electrolyte under OER condition have been completed and supplemented in the revised manuscript (Please see **Fig. S22a**). The dissolved concentration of Cu ions in sample was measured during OER at 10 mA cm^{-2} for every 10 hours. The ICP-MS results showed negligible dissolution of H-Cu₂O after OER-50 hours operation, indicating good stability of the reconstructed surface.

Figure S22. The stability tests of catalysts. (a) Chronopotentiometry tests of H-Cu₂O and the dissolved Cu content for OER at 10 mA cm^{-2} current density after 10, 20-, 30-, 40-, and 50-hours operation.

Q2 : The Cu₂O catalysts have been prepared firstly by anode electrodeposition with the Cu(OH)₂ as the precursor, and the annealing. According to the current demonstration, the as-synthesized Cu₂O would eventually evolve into Cu(OH)₂ again. The authors need to explain why a transformation from Cu(OH)₂ to Cu₂O by annealing is needed. Also, the precursor Cu(OH)₂ should also be one of the control sample.

Reply :

Thank you for reviewer's helpful comments on our manuscript.

According to the referee's sincere and crucial suggestions, the precursor Cu(OH)₂ as the control sample for OER has been measured and supplemented in the revised manuscript (Please see **Fig. 1d**, **Fig. 1e** and **Fig. S9**). Apparently, the OER activity of the precursor Cu(OH)₂ would be worsened greatly due to the low electron conductivity and poor electron-transfer rate in electrochemical reaction process. In this work, the bulk Cu(OH)₂ merely serves as the precursor because of relative ease with which it transforms into Cu₂O. Similar studies in the fabrication of 3d transition-metal-oxide catalysts have also been reported widely in literatures. (Please see ref. by Wang, T. et al., *Nature Catalysis* **2022**, 5 (1), 66-73; Gu, W. et al., *Chemical Communications* **2018**, 54 (90), 12698-12701) However, a large number of surface sites will not be available in the bulk Cu(OH)₂ for electrochemical reactions, limiting their electrochemical performances. It was a coincidence that the dynamic structural reconstruction about forming the surface Cu(OH)₂ (30 nm thickness) was captured during OER (Please see **Fig. S21**). Eventually, the formation of ZR singlet state (i.e. Cu³⁺ with *d⁹L* configuration) exhibited performance on the OER process surprisingly.

Fig. 1 (d) OER polarization curves of Cu₂O, H-Cu₂O, Cu(OH)₂, and Cu foam in KOH solution (1 M).
(e) The corresponding Tafel plots of all catalysts.

Figure S9. EIS Nyquist plots recorded at 1 V vs. RHE in KOH (1 M).

Figure S21. (a) TEM image of H-Cu₂O after OER. SAED patterns of (b) inner bulk and (c) outer surface. (d) Cross section of H-Cu₂O after OER. The enlarged scale image and the corresponding FFT diffraction patterns of frame (e) Cu₂O core region, (f) Cu(OH)₂ reconstruction region and (g) amorphous region from the cross section of H-Cu₂O after OER.

Q3 : In Methods, the description for the synthesis is confusion, such as “surficial Cu(OH)₂ layer on Cu₂O electrodes”, maybe it is on the Cu foam?

Reply :

Thank the referee very much for his/her attention to our carelessness.

The sentence has been corrected carefully in the revised manuscript (Please see yellow-highlight in **Line 349**).

Q4 : In Fig. 5, the structure of CuO_4 square-planer is very confusing, since now it sites outside the OER cycle, which suggest it not one the intermediates for OER. Also now two Cu centers are included in the OER cycle but only one is involved. For the active-site configuration and reaction cycle, more details still need to be double-checked. It is very clear that the CuO_4 square-planer is accompanied with OER, it is very promising to be the active structure. However, there is still some possibility that it is not. To avoid the unnecessary mistake, maybe the author could soften the claims a little bit. No matter what it is, it is still needed to put this CuO_4 square-planer into the OER cycle more properly. If possible, please also try to label all the Cu oxidation state at the OER cycle.

Reply :

Thanks for reviewer's constructive suggestions on our manuscript.

The active-site configuration and reaction cycle in this figure have been corrected in the revised manuscript (Please see **Fig. 5**). The CuO_4 square-planer was put into the OER cycle and to be the intermediate probably. Moreover, now only one Cu center is included in the OER cycle. We further labelled all the possible oxidation state in Cu at the OER cycle. Therefore, the active-site configuration and reaction cycle were repropose.

In this work, many *operando* spectroscopic tools, including GAXS, quick-XAS, soft-XAS, Raman spectra and EIS, were utilized to confirm that such a high-valent CuO_4 geometry containing Cu^{3+} with d^9L charged character is the active structure for OER.

Fig. 5 Proposed OER mechanism for $\text{H-Cu}_2\text{O}$.

Q5 : According Fig. 5, the conventional adsorbed evolution mechanism and intermolecular nucleophilic attack have been proposed for these Cu-based electrocatalysts. If it is the case, then there

should be some special pH-dependence properties. To strengthen these points, the pH-dependence results should be provided.

Reply :

Many thanks for reviewer's helpful comments on our manuscript.

According to the referee's sincere and crucial suggestions, the pH-dependent OER activity has been completed and supplemented in the revised manuscript (Please see **Fig. S23**). Significantly, the H-Cu₂O catalyst exhibits pH-dependent OER activity, implying that non-concerted proton-electron transfers may participate in catalyzing the OER. Furthermore, under alkaline conditions, electrochemically driven deprotonation results in the intramolecular hydroxyl nucleophilic attack pathway where the adjacent OH⁻ attacks CuO₄ to form the O-O bond. Therefore, a pH-dependent nucleophilic attack pathway for O-O bond formation might be proposed (Please see **Fig. S24**). Similar studies in Co-based catalysts have also been reported in literatures. (Please see ref. by Yang, H. et al., *Nature Catalysis* **2022**, 5, 414-429) In order to avoid the confusion for the readers, the active-site configuration and reaction cycle in **Fig. 5** have been corrected thoroughly and discussed in the revised manuscript (Please see **Fig. 5 and Line 322-327**).

Figure S23. LSV curves of H-Cu₂O recorded from pH 12.5 to pH 14.0.

Figure S24. The intramolecular hydroxyl nucleophilic attack process during OER cycle.

Fig. 5 Proposed OER mechanism for H-Cu₂O.

Q6 : In Figure 2d, the surface reconstruction is driven by applied potential, so it is not “self-reconstruction”.

Reply :

Thanks for reviewer’s constructive suggestions on our manuscript.

All of the terms of "self-reconstruction" have been thoroughly replaced with "surface reconstruction" in this revised manuscript (Please see **Line 197, 214, and Figure 2d**).

Fig. 2 (d) Proposed structural reconstruction during OER.

Q7 : In Supplementary Figure 7, it is not a fair comparison since different substrates are used, such herein it is Cu foam, others could be carbon cloth or glassy carbon.

Reply :

Thanks for reviewer’s constructive suggestions on our manuscript.

The comparison of OER catalytic performance has been corrected in the revised manuscript (Please see **Figure S7**). The references using non-metal-foam as substrate were removed thoroughly.

Figure S7. Comparison of OER catalytic performance of H-Cu₂O catalysts with reported Cu-based catalyst and other 3d TM-oxide catalysts.

Reviewer: 2

The authors reported new OER catalyst H-Cu₂O exhibits a superior catalytic activity and stability compared with other Cu-based OER catalysts, which possessed two steps of surface reconstruction from initial catalytic inactive Cu¹⁺ in H-Cu₂O to Cu²⁺ state and further to catalytic active Zhan-Rice singlet state, which is demonstrated as the real active sites for OER rather than CuOOH. This work provides a new insight in understanding of the mechanism of a Cu-based catalyst for water oxidation. Even though the manuscript displayed attractive proofs for Zhang-Rice singlet, some related experiments and discussions are needed to be supplemented before accepted in Nature communications. The detailed comments are as follows:

We thank the reviewer for positive statements and suggestion to improve our manuscripts.

Q1 : In line 100, the authors claimed that the main feature at 933.4 eV in Cu₂O is assigned to final state 1s3d104s1 from initial state 3d10 for Cu¹⁺, while it was pointed out that the feature was originated from 2p3/2p6d10s1 in ref 27. The authors are suggested to confirm the results.

Reply :

Thank the referee very much for his/her attention to our carelessness.

The expression of "final state 1s3d¹⁰4s¹" has been corrected and replaced by "final state 2p_{3/2}p⁶d¹⁰s¹" in the revised manuscript (Please see yellow-highlight in **Line 100**).

Q2 : In line101, the authors claimed that the relative lower intensity at 933.4 eV indicated that the unoccupied states of $4s^1$ increase in H-Cu₂O, which should be explained and demonstrated by some related literature.

Reply :

Many thanks for reviewer's helpful comments on our manuscript.

We also thank the reviewer for the careful reading. In fact, since Cu¹⁺ was fully occupied with ten electrons in $3d$, the broad and weak peak at 933.4 was assigned to the unoccupied of $4s^1$ states in ref 27. The slightly lower intensity of peak at 933.4 eV in H-Cu₂O sample relative to pure Cu₂O indicated the higher electron occupation. Therefore, the unoccupied states of $4s^1$ "decrease" in H-Cu₂O. Herein, the sentence has been corrected carefully in the revised manuscript (Please see yellow-highlight in **Line 101-104**).

Q3 : The oxygen vacancies in Figure 1c should be confirmed by O 1s spectra or electron paramagnetic resonance (EPR) or Positron annihilation lifetime spectrum (PALS). Besides, the existence of oxygen vacancies generally resulted in lower valence state while the Cu in H-Cu₂O still was Cu¹⁺, which is quite weird. The authors are suggested to explain the phenomenon.

Reply :

Thank the referee very much for his/her attention to our typos in **Figure 1c**.

The correct expression is "disorder" instead of "oxygen vacancy" in **Figure 1c**. The **Figure 1c** has been corrected in the revised manuscript (Please see **Fig. 1c**). Such a disorder layer of ~6 nm was confirmed by TEM images as shown in **Fig. S21d-f**. With the great support of *operando* X-ray measurements, the results showed that the structural disorder led to the rapid surface reconstruction at H-Cu₂O during OER. However, in this work, the existence of structural disorder would not change the valence state of Cu. Similar studies in NiO catalysts have also been reported in literatures. (Please see ref. by Fan, L. et al., *ChemSusChem* **2020**, 13, 5901–5909)

Fig. 1 (c) Cu *K*-edge EXAFS spectra of Cu₂O and H-Cu₂O.

Figure S21. (d) Cross section of H-Cu₂O after OER. The enlarged scale image and the corresponding FTT diffraction patterns of frame (e) Cu₂O core region, (f) Cu(OH)₂ reconstruction region and (g) amorphous region from the cross section of H-Cu₂O after OER.

Q4 : LSV curves after 10, 20-, 30-, 40-, and 50-hours durability test at 10 mA cm⁻² should be supplemented to support the stability of the reconstructed surface. Besides, to well evaluate the stability of H-Cu₂O, the durability at 100 mA cm⁻² should be tested for 100 hours.

Reply :

Many thanks for reviewer's helpful comments on our manuscript.

According to the referee's sincere and crucial suggestions, the LSV curves after 10, 20-, 30-, 40-, and 50-hours durability test at 10 mA cm⁻² have been measured and supplemented in the revised manuscript (Please see **Fig. S22**). Besides, the long-term stability tests of catalysts at 100 mA cm⁻² for 100 hours have been completed and supplemented in the revised manuscript (Please see **Fig. S25**). Evidently, the results showed negligible decay of H-Cu₂O after OER-100 hours operation, indicating good stability of the reconstructed surface (Please see yellow-highlight in **Line 133-136**).

Figure S22. The stability tests of catalysts. (a) Chronopotentiometry tests of H-Cu₂O and the dissolved Cu content for OER at 10 mA cm⁻² current density after 10, 20-, 30-, 40-, and 50-hours operation. (b) LSV curves of H-Cu₂O before and after 10, 20-, 30-, 40-, and 50-hours durability test at 10 mA cm⁻².

Figure S25. Extended chronoamperometry measurement at current density of 100 mA cm⁻² for 100 hours.

Q5 : In line 113, the authors held the view that the enhanced localization capacity of charges on Cu atoms probably weakened the chemisorbed oxygen intermediates and improved the water oxidation performance, which should be well discussed the relationship between the localization capacity of charges and oxygen intermediates adsorption. In addition, some related literatures on the binding strength with oxygen intermediates for pure Cu₂O should be well cited and discussed to demonstrate that the weakened chemisorbed oxygen intermediates of Cu₂O are beneficial to OER performance.

Reply :

Thanks for reviewer's constructive suggestions on our manuscript.

According to the referee's sincere and crucial suggestions, the relationship between the localization capacity of charges and oxygen intermediates adsorption has been supplemented in the revised manuscript (Please see yellow-highlight in **Line 116-126**). In addition, the related literatures on the binding strength with oxygen intermediates for Cu-based catalysts have been cited in the revised manuscript (Please see Liang, L. et al., *ACS Applied Nano Materials* **2021**, 4, 6135-6144).

In theory, the bonding strength of oxygen intermediates could depend on the degree of the filling in the antibonding states. The enhanced localization capacity of charges on Cu could contribute to much filling of the antibonding states, resulting in weak adsorption of oxygen intermediates. In contrast, the less filling of the antibonding states would result in the strong adsorption of oxygen intermediates. Similar studies have also been reported in literatures. (Please see ref. by Zhu, K. et al., *Nano Energy* 2020, 73, 104761) According to OER volcano plot in metal oxides, the binding strength with oxygen

intermediates should be neither too strong nor too weak. (Please see ref. by Seh, Z. W. et al., Science 2017, 355, eaad4998)

For p-type Cu_2O , highest occupied d -state is quite closer to Fermi level, resulting in the less filling of antibonding states and the stronger adsorption of oxygen intermediates. Since the adsorption of oxygen on Cu_2O is too strong that restricts OER activity, the H- Cu_2O is effective for much filling of the antibonding states with weak intermediates adsorption, and thus achieving a better activity.

Q6 : CV analysis at 5 mV/s in wide potential range should be added to display the redox peaks of CuI to CuII and CuII to CuIII to support the in-situ generation of CuIII species, which would be consistent with the Figure 2a theoretically.

Reply :

Thanks for reviewer's constructive suggestions on our manuscript.

According to the referee's sincere and crucial suggestions, the CV curves for catalysts at scan rate of 5 mV s^{-1} have been measured and supplemented in the revised manuscript (Please see **Fig. S26**).

During oxidation process, only one anodic peak appeared at low potential and was attributed to the conversion of Cu(I) into Cu(II). Coincidentally, the anodic peak of Cu(II)/Cu(III) oxidation at high potential region highly overlapped the large OER current response. Even so, in the subsequent reduction process on reverse potential scan, the broad cathodic peak was observed and resulted from the reductive transformation of Cu(III) to Cu(II) (Please see yellow-highlight in **Line 271-278**). This is in agreement with the *operando* X-ray measurements.

Figure S26. CV curves of H- Cu_2O , Cu_2O , and $\text{Cu}(\text{OH})_2$ recorded at scan rate of 5 mV s^{-1} .

Q7 : The Tafel region should be reselected in current density between 5 and 10 mA cm^{-2} in Figure 1e.

Reply :

Many thanks for reviewer's helpful comments on our manuscript.

According to the referee's sincere and crucial suggestions, the Tafel region in Figure 1e has been corrected in the revised manuscript (Please see **Fig. 1e**). Furthermore, the smallest Tafel slope of H-Cu₂O was updated with 91 mVdec⁻¹ (Please see **Line 130**).

Fig. 1 (e) The corresponding Tafel plots of all catalysts.

Q8 : In line 191, the authors claimed that “the Cu-Cu bond shows a different amplitude distribution between stages I and II in the k-space from 7.7 to 6.1 Å⁻¹.” However, in Figure 3e at 1.5 V of stage II, the so-called “Cu-Cu bond” possessed almost the same k-space value as Cu-O bond, which should be ascribed to Cu-O bond in second shell. The authors are suggested to Re-discuss the wavelet transform section.

Reply :

Many thanks for reviewer's helpful comments on our manuscript.

In order to clarify the oscillation amplitudes of Cu-Cu bond, the structural parameters obtained from EXAFS fitting for H-Cu₂O, Cu₂O reference, and Cu(OH)₂ reference were supplemented in the revised manuscript (Please see **Fig. S27 and Table S3**). Clearly, the scattering path of Cu-Cu bond at ~3 Å can be obtained for H-Cu₂O, H-Cu₂O (at 1.5 V of stage II), Cu₂O reference, and Cu(OH)₂ reference (Please see **Fig. S13, Fig. S27, Table S1, and Table S3**). This is in agreement with the y-axis of wavelet transform results. Therefore, the possibility of Cu-O bond in second shell has been excluded.

In terms of EXAFS equation (1), the EXAFS signal decays very rapidly with k based on Debye-Waller term in the back-scatterings function. Specifically, a large Debye-Waller factor (σ^2) causes the expression to decrease exponentially over high k of k-space, making EXAFS much weaker over high k as opposed to low-k range. Note that the Cu-Cu bond of Cu₂O, H-Cu₂O, and H-Cu₂O (at 1.5 V of stage II) exhibited a higher σ^2 relative to Cu(OH)₂ (Please see **Table S1 and Table S3**). Therefore, the Cu-Cu bond of Cu₂O, H-Cu₂O, and H-Cu₂O (at 1.5 V of stage II) appeared over low-k range due to the high structural disorder effect. To the sum, we deduce the reason why the Cu-Cu bond of H-Cu₂O at 1.5 V of stage II shifted to low-k region (~6.1 Å⁻¹) relative to H-Cu₂O at 1.4 V of stage I. In order to

avoid the confusion for the readers, two dashed lines at 5.6 and 7.7 Å⁻¹ in **Fig. 3e** have been removed thoroughly in the revised manuscript (Please see **Fig. 3e**).

$$\chi(k) = \sum_j \frac{N_j S_0^2 F_j(k)}{k R_j^2} \exp[-2k^2 \sigma_j^2] \exp\left[\frac{-2R_j}{\lambda(k)}\right] \sin [2k R_j + \phi_j(k)] \dots \dots \dots (1)$$

Figure S13. *Operando* EXAFS spectra of (a) H-Cu₂O.

Figure S27. EXAFS fitting spectra of (a) Cu(OH)₂ reference, (b) Cu₂O reference, and (c) H-Cu₂O.

Table S1. Summary of *Operando* FT-EXAFS fit data for H-Cu₂O.

Sample	Path	CN	R /Å	$\sigma^2/\text{Å}^2$	E_0	R-factor
OCP	Cu-O	1.654	1.87251	0.0092	-5.8596	0.01609
	Cu-Cu	9.96	3.035	0.02618	-8.07442	
1 V	Cu-O	1.682	1.87125	0.00818	-5.43714	0.01317
	Cu-Cu	9.936	3.04226	0.02849	-2.54153	
1.1 V	Cu-O	1.726	1.89978	0.00897	-0.40354	0.01370
	Cu-Cu	9.48	2.96973	0.02800	-7.69081	
1.2 V	Cu-O	1.782	1.9126	0.00852	-0.39692	0.01879
	Cu-Cu	6.084	2.98916	0.02301	-6.11109	
1.3 V	Cu-O	2.146	1.93434	0.00896	9.01099	0.01829
	Cu-Cu	2.34	2.92481	0.01224	-2.89101	
1.4 V	Cu-O	2.326	1.95172	0.00893	9.552315	0.01995
	Cu-Cu	2.424	2.90807	0.01243	-0.277395	
1.5 V	Cu-O	1.954	1.93857	0.00592	-2.52865	0.01848
	Cu-Cu	2.376	2.90807	0.01233	14.39303	

Table S3. EXAFS fitting results for the structural parameters around Cu atoms.

sample	Path	CN	R /Å	$\sigma^2(\text{Å}^2)$	E_0	R-factor
Cu ₂ O Reference	Cu-O	2	1.85207	0.00524	8.421	0.01663
	Cu-Cu	12	3.03349	0.02132	8.075	
Cu(OH) ₂ Reference	Cu-O	4	1.95044	0.00599	-2.065	0.00729
	Cu-O	1	2.55788	0.00699	14.389	
	Cu-Cu	2.84	3.06211	0.00620	17.882	
H-Cu ₂ O	Cu-O	1.637	1.86251	0.00845	-3.8578	0.01749
	Cu-Cu	9.74	3.04512	0.02418	-6.07742	

Fig. 3 (e) Comparison of Cu K -edge WT-EXAFS recorded for H-Cu₂O, standard references and catalytic materials at OCP, 1.4, 1.5 V and after OER.

Q9 : It should be stated clearly that the data of “during OER” in Figure 4b and 4c was collected at what OER potential and more XAS data at different OER potentials such as 1.6 V and 1.8 V should be supplemented to support the Zhang-Rice singlet was the active sites for OER in H-Cu₂O.

Reply :

Thanks for reviewer’s constructive suggestions on our manuscript.

In this work, the OER potential at 1.5 V was termed "during OER."

In order to avoid the confusion for the readers, we have replaced the expression of "during OER" by "1.5 V" in the revised manuscript (Please see Fig. 4b, Fig. 4c, Fig. S15, Fig. S16, and Fig. S17). Furthermore, the soft-XAS data at different OER potentials have been supplemented systematically in the revised manuscript (Please see Fig. 4b, Fig. 4c, Fig. S15, Fig. S16, and Fig. S17). Evidently, the Zhang-Rice singlet on H-Cu₂O appeared under an applied voltage above 1.5 V.

Fig. 4 (b) Cu L_3 -edge and (c) O K -edge of H-Cu₂O.

Figure S15. Cu L_3 -edge soft-XAS spectra. (a) H-Cu₂O at 1.4 V and 1.5 V, (b) Cu³⁺ in La_{1-x}Sr_xCuO₄ from reference¹⁹.

Figure S16. O *K*-edge soft-XAS spectra. (a) H-Cu₂O during OER, (b) La_{2-x}Sr_xCuO₄ (from ref.²⁵) as Cu³⁺ reference, (c) Sr₁₄Cu₂₄O₄₁ containing two Cu³⁺: CuO₂ chains and leg Cu₂O₃ ladders (from ref.²⁶).

Figure S17. O *K*-edge fit of H-Cu₂O during OER. The dashed blue line represents the pure corner-shared Cu-O network; the green line represents the pure edge-shared Cu-O network.

Q10 : The Zhang-rice singlet was observed in H-Cu₂O as the reconstruction surface Cu(OH)₂ generation. Herein, it should be pointed out whether Zhang-rice singlet can be observed for pure Cu(OH)₂ or pure Cu₂O.

Reply :

Thanks for reviewer's constructive suggestions on our manuscript.

In order to make sure whether Zhang-rice singlet can be observed for pure Cu(OH)₂ or pure Cu₂O, the CV curves for pure Cu(OH)₂ and pure Cu₂O at scan rate of 5 mV s⁻¹ have been measured and supplemented in the revised manuscript (Please see **Fig. S26**).

During oxidation process, only one anodic peak appeared at low potential and was attributed to the conversion of Cu(I) into Cu(II). Coincidentally, the anodic peak of Cu(II)/Cu(III) oxidation at high potential region highly overlapped the large OER current response. Even so, in the subsequent reduction process on reverse potential scan, the broad cathodic peak was observed and resulted from the reductive transformation of Cu(III) to Cu(II) (Please see **Line 271-278**). Note that, relative to H-Cu₂O, the Zhang-rice singlet (Cu³⁺) can also be observed for pure Cu(OH)₂ and pure Cu₂O although they appeared in much high potential region due to the poor OER kinetics.

Figure S26. CV curves of H-Cu₂O, Cu₂O, and Cu(OH)₂ recorded at scan rate of 5 mV s⁻¹.

Q11 : The improved activity of H-Cu₂O was ascribed to the formation of disordered layer compared with pure Cu₂O. However, the activity comparison between H-Cu₂O and pure Cu₂O after durability test at 100 mA cm⁻² for 100 hours should be supplemented since both H-Cu₂O and pure Cu₂O finally reconstructed Cu(OH)₂ on the surface.

Reply :

Thanks for reviewer's constructive suggestions on our manuscript.

According to the referee's sincere and crucial suggestions, the activity comparison between H-Cu₂O and pure Cu₂O after durability test at 100 mA cm⁻² have been completed and supplemented in the revised manuscript (Please see **Fig. S28**).

As illustrated in **Fig. S28**, the H-Cu₂O exhibited a positive shift of 67 mV at a current density of 100 mA cm⁻² for 100 hours, while the pure Cu₂O demonstrated 114 mV positive shifts at a current density of 100 mA cm⁻² for 65 hours. Relative to pure Cu₂O, the H-Cu₂O showed good stability and activity.

Figure S28. Polarization curves of H-Cu₂O and Cu₂O were recorded before and after durability test at 100 mA cm⁻².

Reviewer: 3

The manuscript by Peng et al. provides a thorough study of Cu-based catalysts for the oxygen evolution reaction (OER). In spite of the vast variety of *operando*, in situ and ex situ methods, the analysis and discussion of the obtained results is not fully provided thus it is hard for a reader to follow the logic and make any conclusions based on the presented information. First of all, I suggest that this work is published as a full paper (not Nature Communication) where the authors should properly discuss all their results.

We thank the reviewer for positive statements and suggestions to improve our manuscripts. Currently, the various *operando* experiments are essential and necessary for the high-level journal. In this work, we focused on the electronic structure by utilizing the soft-XAS. For the atomic structure, we used the hard-XAS for probing geometry sites. Most importantly, many *operando* spectroscopic tools, including grazing-angle X-ray scattering (GAXS), quick X-ray absorption (quick-XAS), soft X-ray absorption (soft-XAS), Raman spectra and electrochemical impedance spectroscopy (EIS), were utilized to uncover that the Zhang-Rice (ZR) singlet state is unexpectedly observed to participate directly in OER-cycle superconductors. Furthermore, we are quite glad to answer the questions from the reviewer one by one.

Secondly, the following changes are suggested to the authors:

Q1 : The authors included a lot of data in the graphs which are hard to read and analyze. E.g. XANES data contain all the measured potentials. It would be more beneficial to select the critical potentials and plot them together for Cu₂O and H-Cu₂O (where the full set remains in SI). Moreover, please add the XANES spectra for the reference samples.

Reply :

Many thanks for reviewer's helpful comments on our manuscript.

According to the referee's sincere and crucial suggestions, we have selected the XANES under several critical potentials and further plotted them together for Cu₂O, H-Cu₂O, and reference samples in the revised manuscript (Please see **Fig. S29**). The detailed discussions of the *operando* XANES for Cu *K*-edge of H-Cu₂O and Cu₂O were shown in the revised manuscript (Please see **Line 179-183**).

Figure S29. Selective *operando* XANES for Cu K-edge of (a) H-Cu₂O and (b) pure Cu₂O together with the reference samples.

Q2 : The discussion of Cu K edge spectra (as well as other techniques, see my main comment above) is very limited. The authors discuss only the edge shift and its correlation with oxidation state. The discussion of the pre-edge region is absent, where this part of the XANES spectra contains vital information on the local coordination and should be correlated with EXAFS and Cu L edge presented later. The authors claim the formation of square planar Cu sites under OER conditions, however, this should be clearly seen in the Cu K edge spectra (pre-edge region, see e.g. <https://pubs.acs.org/doi/10.1021/acs.jpcclett.8b00675>). The conclusions on the geometry changes upon the reaction should be reconsidered based on the available data.

Reply :

Thanks for reviewer's constructive suggestions on our manuscript.

According to the referee's sincere and crucial suggestions, more detailed discussions of the pre-edge region in the Cu K-edge spectra have been completed and supplemented in the revised manuscript (Please see yellow-highlight in **Line 179-183, 200, and 229**).

As shown in **Fig 3a, b**, the appearance of a slight shoulder at ~ 8985 eV on the rising K-edge XAS at applied voltage 1.5 V for H-Cu₂O and 1.7 V for Cu₂O, respectively. The feature can be assigned to four-coordinated square-planar geometry (Please see ref. by Zhang, R. et al., *The Journal of Physical Chemistry Letters* **2018**, 9, 3035-3042), which completely agrees with the *operando* Cu L₃-edge results (Please see **Fig. 4b**). On this basis, the oxidation state of the copper site at stage II might be assigned as +3, showing the electrochemically driven conversion of Cu(OH)₂ into CuO₄ geometry. This is also positively correlated to the decreased $\Delta N/N_{\text{OCP}}$ and decreased $\Delta R/R_{\text{OCP}}$ of the catalysts at stage II, as probed by EXAFS (Please see **Fig 3c, d**).

Fig. 3 (a, b) 3D patterns of *operando* Cu K-edge XANES of H-Cu₂O and Cu₂O catalysts. (c, d) Structural coherence change of Cu-O in EXAFS coordination number and bond length of H-Cu₂O and Cu₂O catalysts under an applied potential relative to the OCP state.

Fig. 4 *Operando* soft-XAS characterization of high-valent Cu as active site during OER. (b) Cu L₃-edge of H-Cu₂O.

Q3 : Please add Cu L edge data recorded on Cu₂O sample (in situ if possible) and the references for comparison.

Reply :

Many thanks for reviewer's helpful comments on our manuscript.

According to the referee's sincere and crucial suggestions, the Cu *L*-edge data recorded on Cu₂O samples and the references have been completed and supplemented in the revised manuscript (Please see yellow-highlight in **Line 101**). Clearly, both of Cu₂O and H-Cu₂O confirmed the presence of Cu¹⁺ surface.

Figure 1. (b) Cu *L*₃-edge XANES spectra of Cu₂O, H-Cu₂O, and the reference samples.

Thank you again for the critical but helpful comments from the reviewers. We have tried our best to revise the manuscript, making it more scientific and acceptable for journal standard and general readers.

REVIEWERS' COMMENTS

Reviewer #1 (Remarks to the Author):

I appreciate the efforts from the authors to address all the raised problems.

Reviewer #2 (Remarks to the Author):

I am satisfied with the revision. I recommend this manuscript to be published in Nature Communications.

Reviewer #3 (Remarks to the Author):

The authors revised the manuscript according to the reviewers remarks, the publication has been improved. It can be accepted to Nature Communications.

Reply to Reviewers

Reviewer #1 (Remarks to the Author):

I appreciate the efforts from the authors to address all the raised problems.

Reviewer #2 (Remarks to the Author):

I am satisfied with the revision. I recommend this manuscript to be published in Nature Communications.

Reviewer #3 (Remarks to the Author):

The authors revised the manuscript according to the reviewers remarks, the publication has been improved. It can be accepted to Nature Communications.

We would like to thank Reviewer #1, Reviewer #2 and Reviewer #3 for recommending our revised manuscript to be published in *Nature Communications*.